# Robust Logistic Regression and Classification

**Jiashi Feng**
EECS Department & ICSI
UC Berkeley
jshfeng@berkeley.edu

**Huan Xu**
ME Department
National University of Singapore
mpexuh@nus.edu.sg

**Shie Mannor**
EE Department
Technion
shie@ee.technion.ac.il

**Shuicheng Yan**
ECE Department
National University of Singapore
eleyans@nus.edu.sg

## Abstract

We consider logistic regression with arbitrary outliers in the covariate matrix. We propose a new robust logistic regression algorithm, called RoLR, that estimates the parameter through a simple linear programming procedure. We prove that RoLR is robust to a constant fraction of adversarial outliers. To the best of our knowledge, this is the first result on estimating logistic regression model when the covariate matrix is corrupted with any performance guarantees. Besides regression, we apply RoLR to solving binary classification problems where a fraction of training samples are corrupted.

## 1   Introduction

Logistic regression (LR) is a standard probabilistic statistical classification model that has been extensively used across disciplines such as computer vision, marketing, social sciences, to name a few. Different from linear regression, the outcome of LR on one sample is the *probability* that it is positive or negative, where the probability depends on a linear measure of the sample. Therefore, LR is actually widely used for classification. More formally, for a sample $x_i \in \mathbb{R}^p$ whose label is denoted as $y_i$, the probability of $y_i$ being positive is predicted to be $\mathbb{P}\{y_i = +1\} = \frac{1}{1+e^{-\beta^\top x_i}}$, given the LR model parameter $\beta$. In order to obtain a parameter that performs well, often a set of labeled samples $\{(x_1, y_1), \ldots, (x_n, y_n)\}$ are collected to learn the LR parameter $\beta$ which maximizes the induced likelihood function over the training samples.

However, in practice, the training samples $x_1, \ldots, x_n$ are usually noisy and some of them may even contain *adversarial corruptions*. Here by "adversarial", we mean that the corruptions can be arbitrary, unbounded and are not from any specific distribution. For example, in the image/video classification task, some images or videos may be corrupted unexpectedly due to the error of sensors or the severe occlusions on the contained objects. Those corrupted samples, which are called *outliers*, can skew the parameter estimation severely and hence destroy the performance of LR.

To see the sensitiveness of LR to outliers more intuitively, consider a simple example where all the samples $x_i$'s are from one-dimensional space $\mathbb{R}$, as shown in Figure 1. Only using the inlier samples provides a correct LR parameter (we here show the induced function curve) which explains the inliers well. However, when only one sample is corrupted (which is originally negative but now closer to the positive samples), the resulted regression curve is distracted far away from the ground truth one and the label predictions on the concerned inliers are completely wrong. This demonstrates that LR is indeed fragile to sample corruptions. More rigorously, the non-robustness of LR can be shown via calculating its influence function [7] (detailed in the supplementary material).

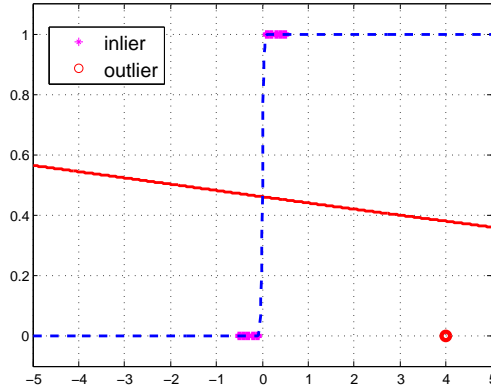

Figure 1: The estimated logistic regression curve (red solid) is far away from the correct one (blue dashed) due to the existence of just one outlier (red circle).

As Figure 1 demonstrates, the maximal-likelihood estimate of LR is extremely sensitive to the presence of anomalous data in the sample. Pregibon also observed this non-robustness of LR in [14]. To solve this important issue of LR, Pregibon [14], Cook and Weisberg [4] and Johnson [9] proposed procedures to identify observations which are influential for estimating $\beta$ based on certain outlyingness measure. Stefanski *et al.* [16, 10] and Bianco *et al.* [2] also proposed robust estimators which, however, require to robustly estimating the covariate matrix or boundedness on the outliers. Moreover, the breakdown point[1] of those methods is generally inversely proportional to the sample dimensionality and diminishes rapidly for high-dimensional samples.

We propose a new robust logistic regression algorithm, called RoLR, which optimizes a *robustified* linear correlation between response $y$ and linear measure $\langle \beta, x \rangle$ via an efficient linear programming-based procedure. We demonstrate that the proposed RoLR achieves robustness to arbitrarily covariate corruptions. Even when a constant fraction of the training samples are corrupted, RoLR is still able to learn the LR parameter with a non-trivial upper bound on the error. Besides this theoretical guarantee of RoLR on the parameter estimation, we also provide the empirical and population risks bounds for RoLR. Moreover, RoLR only needs to solve a linear programming problem and thus is scalable to large-scale data sets, in sharp contrast to previous LR optimization algorithms which typically resort to (computationally expensive) iterative reweighted method [11]. The proposed RoLR can be easily adapted to solving binary classification problems where corrupted training samples are present. We also provide theoretical classification performance guarantee for RoLR. Due to the space limitation, we defer all the proofs to the supplementary material.

## 2   Related Works

Several previous works have investigated multiple approaches to robustify the logistic regression (LR) [15, 13, 17, 16, 10]. The majority of them are M-estimator based: minimizing a complicated and more robust loss function than the standard loss function (negative log-likelihood) of LR. For example, Pregiobon [15] proposed the following M-estimator:

$$\hat{\beta} = \arg\min_{\beta} \sum_{i=1}^{n} \rho(\ell_i(\beta)),$$

where $\ell_i(\cdot)$ is the negative log-likelihood of the $i$th sample $x_i$ and $\rho(\cdot)$ is a Huber type function [8] such as

$$\rho(t) = \begin{cases} t, & \text{if } t \le c, \\ 2\sqrt{tc} - c, & \text{if } t > c, \end{cases}$$

with $c$ a positive parameter. However, the result from such estimator is not robust to outliers with high leverage covariates as shown in [5].

Recently, Ding *et al* [6] introduced the $T$-logistic regression as a robust alternative to the standard LR, which replaces the exponential distribution in LR by $t$-exponential distribution family. However, $T$-logistic regression only guarantees that the output parameter converges to a local optimum of the loss function instead of converging to the ground truth parameter.

Our work is largely inspired by following two recent works [3, 13] on robust sparse regression. In [3], Chen *et al.* proposed to replace the standard vector inner product by a trimmed one, and obtained a novel linear regression algorithm which is robust to unbounded covariate corruptions. In this work, we also utilize this simple yet powerful operation to achieve robustness. In [13], a convex programming method for estimating the sparse parameters of logistic regression model is proposed:

$$\max_{\beta} \sum_{i=1}^{m} y_i \langle x_i, \beta \rangle, \text{ s.t. } \|\beta\|_1 \leq \sqrt{s}, \|\beta\| \leq 1,$$

where $s$ is the sparseness prior parameter on $\beta$. However, this method is not robust to corrupted covariate matrix. Few or even one corrupted sample may dominate the correlation in the objective function and yield arbitrarily bad estimations. In this work, we propose a robust algorithm to remedy this issue.

## 3 Robust Logistic Regression

### 3.1 Problem Setup

We consider the problem of logistic regression (LR). Let $S^{p-1}$ denote the unit sphere and $B_2^p$ denote the Euclidean unit ball in $\mathbb{R}^p$. Let $\beta^*$ be the groundtruth parameter of the LR model. We assume the training samples are covariate-response pairs $\{(x_i, y_i)\}_{i=1}^{n+n_1} \subset \mathbb{R}^p \times \{-1, +1\}$, which, if not corrupted, would obey the following LR model:

$$\mathbb{P}\{y_i = +1\} = \tau(\langle \beta^*, x_i \rangle + v_i), \tag{1}$$

where the function $\tau(\cdot)$ is defined as: $\tau(z) = \frac{1}{1+e^{-z}}$. The additive noise $v_i \sim \mathcal{N}(0, \sigma_e^2)$ is an i.i.d. Gaussian random variable with zero mean and variance of $\sigma_e^2$. In particular, when we consider the noiseless case, we assume $\sigma_e^2 = 0$. Since LR only depends on $\langle \beta^*, x_i \rangle$, we can always scale the samples $x_i$ to make the magnitude of $\beta^*$ less than 1. Thus, without loss of generality, we assume that $\beta^* \in S^{p-1}$.

Out of the $n + n_1$ samples, a constant number $(n_1)$ of the samples may be adversarially corrupted, and we make no assumptions on these outliers. Throughout the paper, we use $\lambda \triangleq \frac{n_1}{n}$ to denote the outlier fraction. We call the remaining $n$ non-corrupted samples "authentic" samples, which obey the following standard sub-Gaussian design [12, 3].

**Definition 1** (Sub-Gaussian design). *We say that a random matrix $X = [x_1, \ldots, x_n] \in \mathbb{R}^{p \times n}$ is sub-Gaussian with parameter $(\frac{1}{n}\Sigma_x, \frac{1}{n}\sigma_x^2)$ if: (1) each column $x_i \in \mathbb{R}^p$ is sampled independently from a zero-mean distribution with covariance $\frac{1}{n}\Sigma_x$, and (2) for any unit vector $u \in \mathbb{R}^p$, the random variable $u^\top x_i$ is sub-Gaussian with parameter[2] $\frac{1}{\sqrt{n}}\sigma_x$.*

The above sub-Gaussian random variables have several nice concentration properties, one of which is stated in the following Lemma [12].

**Lemma 1** (Sub-Gaussian Concentration [12]). *Let $X_1, \ldots, X_n$ be $n$ i.i.d. zero-mean sub-Gaussian random variables with parameter $\sigma_x/\sqrt{n}$ and variance at most $\sigma_x^2/n$. Then we have $\left|\sum_{i=1}^{n} X_i^2 - \sigma_x^2\right| \leq c_1 \sigma_x^2 \sqrt{\frac{\log p}{n}}$, with probability of at least $1 - p^{-2}$ for some absolute constant $c_1$.*

Based on the above concentration property, we can obtain following bound on the magnitude of a collection of sub-Gaussian random variables [3].

**Lemma 2.** *Suppose $X_1, \ldots, X_n$ are $n$ independent sub-Gaussian random variables with parameter $\sigma_x/\sqrt{n}$. Then we have $\max_{i=1,\ldots,n} |X_i| \leq 4\sigma_x \sqrt{(\log n + \log p)/n}$ with probability of at least $1 - p^{-2}$.*

Also, this lemma provides a rough bound on the magnitude of inlier samples, and this bound serves as a threshold for pre-processing the samples in the following RoLR algorithm.

## 3.2 RoLR Algorithm

We now proceed to introduce the details of the proposed Robust Logistic Regression (RoLR) algorithm. Basically, RoLR first removes the samples with overly large magnitude and then maximizes a trimmed correlation of the remained samples with the estimated LR model. The intuition behind the RoLR maximizing the trimmed correlation is: if the outliers have too large magnitude, they will not contribute to the correlation and thus not affect the LR parameter learning. Otherwise, they have bounded affect on the LR learning (which actually can be bounded by the inlier samples due to our adopting the trimmed statistic). Algorithm 1 gives the implementation details of RoLR.

---

**Algorithm 1** RoLR

**Input**: Contaminated training samples $\{(x_1, y_1), \ldots, (x_{n+n_1}, y_{n+n_1})\}$, an upper bound on the number of outliers $n_1$, number of inliers $n$ and sample dimension $p$.
**Initialization**: Set $T = 4\sqrt{\log p/n + \log n/n}$.
**Preprocessing**: Remove samples $(x_i, y_i)$ whose magnitude satisfies $\|x_i\| \geq T$.
Solve the following linear programming problem (see Eqn. (3)):

$$\hat{\beta} = \arg\max_{\beta \in B_2^p} \sum_{i=1}^{n} [y\langle\beta, x\rangle]_{(i)}.$$

**Output**: $\hat{\beta}$.

---

Note that, within the RoLR algorithm, we need to optimize the following sorted statistic:

$$\max_{\beta \in B_2^p} \sum_{i=1}^{n} [y\langle\beta, x\rangle]_{(i)}. \tag{2}$$

where $[\cdot]_{(i)}$ is a sorted statistic such that $[z]_{(1)} \leq [z]_{(2)} \leq \ldots \leq [z]_{(n)}$, and $z$ denotes the involved variable. The problem in Eqn. (2) is equivalent to minimizing the summation of top $n$ variables, which is a convex one and can be solved by an off-the-shelf solver (such as CVX). Here, we note that it can also be converted to the following linear programming problem (with a quadratic constraint), which enjoys higher computational efficiency. To see this, we first introduce auxiliary variables $t_i \in \{0, 1\}$ as indicators of whether the corresponding terms $y_i\langle\beta, -x_i\rangle$ fall in the smallest $n$ ones. Then, we write the problem in Eqn. (2) as

$$\max_{\beta \in B_2^p} \min_{t_i} \sum_{i=1}^{n+n_1} t_i \cdot y_i\langle\beta, x_i\rangle, \text{ s.t. } \sum_{i=1}^{n+n_1} t_i \leq n, 0 \leq t_i \leq 1.$$

Here the constraints of $\sum_{i=1}^{n+n_1} t_i \leq n, 0 \leq t_i \leq 1$ are from standard reformulation of $\sum_{i=1}^{n+n_1} t_i = n, t_i \in \{0, 1\}$. Now, the above problem becomes a max-min linear programming. To decouple the variables $\beta$ and $t_i$, we turn to solving the dual form of the inner minimization problem. Let $\nu$, and $\xi_i$ be the Lagrange multipliers for the constraints $\sum_{i=1}^{n+n_1} t_i \leq n$ and $t_i \leq 1$ respectively. Then the dual form w.r.t. $t_i$ of the above problem is:

$$\max_{\beta, \nu, \xi_i} -\nu \cdot n - \sum_{i=1}^{n+n_1} \xi_i, \text{ s.t. } y_i\langle\beta, x_i\rangle + \nu + \xi_i \geq 0, \beta \in B_2^p, \nu \geq 0, \xi_i \geq 0. \tag{3}$$

Reformulating logistic regression into a linear programming problem as above significantly enhances the scalability of LR in handling large-scale datasets, a property very appealing in practice, since linear programming is known to be computationally efficient and has no problem dealing with up to $1 \times 10^6$ variables in a standard PC.

## 3.3 Performance Guarantee for RoLR

In contrast to traditional LR algorithms, RoLR does not perform a maximal likelihood estimation. Instead, RoLR maximizes the correlation $y_i\langle\beta, x_i\rangle$. This strategy reduces the computational complexity of LR, and more importantly enhances the robustness of the parameter estimation, using

the fact that the authentic samples usually have positive correlation between the $y_i$ and $\langle \beta, x_i \rangle$, as described in the following lemma.

**Lemma 3.** *Fix $\beta \in S^{p-1}$. Suppose that the sample $(x, y)$ is generated by the model described in* (1). *The expectation of the product $y\langle \beta, x \rangle$ is computed as:*

$$\mathbb{E} y \langle \beta, x \rangle = \mathbb{E} \operatorname{sech}^2(g/2),$$

*where $g \in \mathcal{N}(0, \sigma_x^2 + \sigma_e^2)$ is a Gaussian random variable and $\sigma_e^2$ is the noise level in* (1). *Furthermore, the above expectation can be bounded as follows,*

$$\varphi^+(\sigma_e^2, \sigma_x^2) \leq \mathbb{E} y \langle \beta, x \rangle \leq \varphi^-(\sigma_e^2, \sigma_x^2).$$

*where $\varphi^+(\sigma_e^2, \sigma_x^2)$ and $\varphi^-(\sigma_e^2, \sigma_x^2)$ are positive. In particular, they can take the form of $\varphi^+(\sigma_e^2, \sigma_x^2) = \frac{\sigma_x^2}{3} \operatorname{sech}^2 \left( \frac{1+\sigma_e^2}{2} \right)$ and $\varphi^-(\sigma_e^2, \sigma_x^2) = \frac{\sigma_x^2}{3} + \frac{\sigma_x^2}{6} \operatorname{sech}^2 \left( \frac{1+\sigma_e^2}{2} \right).$*

The following lemma shows the difference of correlations is an effective surrogate for the difference of the LR parameters. Thus we can always minimize the difference of $\|\hat{\beta} - \beta^*\|$ through maximizing $\sum_i y_i \langle \hat{\beta}, x_i \rangle$.

**Lemma 4.** *Fix $\beta \in S^{p-1}$ as the groundtruth parameter in* (1) *and $\beta' \in B_2^p$. Denote $\eta = \mathbb{E} y \langle \beta, x \rangle$. Then*

$$\mathbb{E} y \langle \beta', x \rangle = \eta \langle \beta, \beta' \rangle,$$

*and thus,*

$$\mathbb{E} \left[ y \langle \beta, x \rangle - y \langle \beta', x \rangle \right] = \eta (1 - \langle \beta, \beta' \rangle) \geq \frac{\eta}{2} \| \beta - \beta' \|_2^2.$$

Based on these two lemmas, along with some concentration properties of the inlier samples (shown in the supplementary material), we have the following performance guarantee of RoLR on LR model parameter recovery.

**Theorem 1** (RoLR for recovering LR parameter). *Let $\lambda \triangleq \frac{n_1}{n}$ be the outlier fraction, $\hat{\beta}$ be the output of Algorithm 1, and $\beta^*$ be the ground truth parameter. Suppose that there are $n$ authentic samples generated by the model described in* (1). *Then we have, with probability larger than $1 - 4\exp(-c_2 n/8)$,*

$$\|\hat{\beta} - \beta^*\| \leq 2\lambda \frac{\varphi^-(\sigma_e^2, \sigma_x^2)}{\varphi^+(\sigma_e^2, \sigma_x^2)} + \frac{2(\lambda + 4 + 5\sqrt{\lambda})}{\varphi^+(\sigma_e^2, \sigma_x^2)} \sqrt{\frac{p}{n}} + \frac{8\lambda}{\varphi^+(\sigma_e^2, \sigma_x^2)} \sigma_x^2 \sqrt{\frac{\log p}{n} + \frac{\log n}{n}}.$$

*Here $c_2$ is an absolute constant.*

**Remark 1.** *To make the above results more explicit, we consider the asymptotic case where $p/n \to 0$. Thus the above bounds become*

$$\|\hat{\beta} - \beta^*\| \leq 2\lambda \frac{\varphi^-(\sigma_e^2, \sigma_x^2)}{\varphi^+(\sigma_e^2, \sigma_x^2)},$$

*which holds with probability larger than $1 - 4\exp(-c_2 n/8)$. In the noiseless case, i.e., $\sigma_e = 0$, and assuming $\sigma_x^2 = 1$, we have $\varphi^+(\sigma_e^2) = \frac{1}{3} \operatorname{sech}^2 \left( \frac{1}{2} \right) \approx 0.2622$ and $\varphi^-(\sigma_e^2 + 1) = \frac{1}{3} + \frac{1}{6} \operatorname{sech}^2 \left( \frac{1}{2} \right) \approx 0.4644$. The ratio is $\varphi^-/\varphi^+ \approx 1.7715$. Thus the bound is simplified to:*

$$\|\hat{\beta} - \beta^*\| \lesssim 3.54\lambda.$$

*Recall that $\hat{\beta}, \beta^* \in S^{p-1}$ and the maximal value of $\|\hat{\beta} - \beta^*\|$ is 2. Thus, for the above result to be non-trivial, we need $3.54\lambda \leq 2$, namely $\lambda \leq 0.56$. In other words, in the noiseless case, the RoLR is able to estimate the LR parameter with a non-trivial error bound (also known as a "breakdown point") with up to $0.56/1.56 \times 100\% = 36\%$ of the samples being outliers.*

## 4 Empirical and Population Risk Bounds of RoLR

Besides the parameter recovery, we are also concerned about the prediction performance of the estimated LR model in practice. The standard prediction loss function $\ell(\cdot, \cdot)$ of LR is a non-negative and bounded function, and is defined as:

$$\ell((x_i, y_i), \beta) = \frac{1}{1 + \exp\{-y_i \beta^\top x_i\}}. \tag{4}$$

The goodness of an LR predictor $\beta$ is measured by its *population risk*:

$$R(\beta) = \mathbb{E}_{P(X,Y)} \ell((x,y),\beta),$$

where $P(X,Y)$ describes the joint distribution of covariate $X$ and response $Y$. However, the population risk rarely can be calculated directly as the distribution $P(X,Y)$ is usually unknown. In practice, we often consider the *empirical risk*, which is calculated over the provided training samples as follows:

$$R_{\text{emp}}(\beta) = \frac{1}{n} \sum_{i=1}^{n} \ell((x_i, y_i), \beta).$$

Note that the empirical risk is computed only over the authentic samples, hence cannot be directly optimized when outliers exist.

Based on the bound of $\|\hat{\beta} - \beta^*\|$ provided in Theorem 1, we can easily obtain the following empirical risk bound for RoLR as the LR loss function given in Eqn. (4) is Lipschitz continuous.

**Corollary 1** (Bound on the empirical risk). *Let $\hat{\beta}$ be the output of Algorithm 1, and $\beta^*$ be the optimal parameter minimizing the empirical risk. Suppose that there are $n$ authentic samples generated by the model described in* (1). *Define $X \triangleq 4\sigma_x \sqrt{(\log n + \log p)/n}$. Then we have, with probability larger than $1 - 4\exp(-c_2 n/8)$, the empirical risk of $\hat{\beta}$ is bounded by,*

$$R_{\text{emp}}(\hat{\beta}) - R_{\text{emp}}(\beta^*) \leq \quad X \left\{ 2\lambda \frac{\varphi^-(\sigma_e^2, \sigma_x^2)}{\varphi^+(\sigma_e^2, \sigma_x^2)} + \frac{2(\lambda + 4 + 5\sqrt{\lambda})}{\varphi^+(\sigma_e^2, \sigma_x^2)} \sqrt{\frac{p}{n}} \right.$$

$$\left. + \frac{8\lambda \sigma_x^2}{\varphi^+(\sigma_e^2, \sigma_x^2)} \sqrt{\frac{\log p}{n} + \frac{\log n}{n}} \right\}.$$

Given the empirical risk bound, we can readily obtain the bound on the population risk by referring to standard generalization results in terms of various function class complexities. Some widely used complexity measures include the VC-dimension [18] and the Rademacher and Gaussian complexity [1]. Compared with the Rademacher complexity which is data dependent, the VC-dimension is more universal although the resulting generalization bound can be slightly loose. Here, we adopt the VC-dimension to measure the function complexity and obtain the following population risk bound.

**Corollary 2** (Bound on the population risk). *Let $\hat{\beta}$ be the output of Algorithm 1, and $\beta^*$ be the optimal parameter. Suppose the parameter space $S^{p-1} \ni \beta$ has finite VC dimension $d$. There are $n$ authentic samples are generated by the model described in* (1). *Define $X \triangleq 4\sigma_x \sqrt{(\log n + \log p)/n}$. Then we have, with high probability larger larger than $1 - 4\exp(-c_2 n/8) - \delta$, the population risk of $\hat{\beta}$ is bounded by,*

$$R(\hat{\beta}) - R(\beta^*) \leq X \left\{ 2\lambda \frac{\varphi^-(\sigma_e^2, \sigma_x^2)}{\varphi^+(\sigma_e^2, \sigma_x^2)} + \frac{2(\lambda + 4 + 5\sqrt{\lambda})}{\varphi^+(\sigma_e^2, \sigma_x^2)} \sqrt{\frac{p}{n}} + \frac{8\lambda \sigma_x^2}{\varphi^+(\sigma_e^2, \sigma_x^2)} \sqrt{\frac{\log p}{n} + \frac{\log n}{n}} \right.$$

$$\left. + 2c_3 \sqrt{\frac{d + \ln(1/\delta)}{n}} \right\}.$$

*Here both $c_2$ and $c_3$ are absolute constants.*

## 5 Robust Binary Classification

### 5.1 Problem Setup

Different from the sample generation model for LR, in the standard binary classification setting, the label $y_i$ of a sample $x_i$ is *deterministically* determined by the sign of the linear measure of the sample $\langle \beta^*, x_i \rangle$. Namely, the samples are generated by the following model:

$$y_i = \text{sign}\left(\langle \beta^*, x_i \rangle + v_i\right). \tag{5}$$

Here $v_i$ is a Gaussian noise as in Eqn. (1). Since $y_i$ is deterministically related to $\langle \beta^*, x_i \rangle$, the expected correlation $\mathbb{E} y \langle \beta, x \rangle$ achieves the maximal value in this setup (ref. Lemma 5), which ensures that the RoLR also performs well for classification. We again assume that the training samples contain $n$ authentic samples and at most $n_1$ outliers.

## 5.2 Performance Guarantee for Robust Classification

**Lemma 5.** *Fix $\beta \in S^{p-1}$. Suppose the sample $(x, y)$ is generated by the model described in* (5). *The expectation of the product $y\langle \beta, x \rangle$ is computed as:*

$$\mathbb{E}y\langle \beta, x \rangle = \sqrt{\frac{2\sigma_x^4}{\pi(\sigma_x^2 + \sigma_v^2)}}.$$

Comparing the above result with the one in Lemma 3, here for the binary classification, we can exactly calculate the expectation of the correlation, and this expectation is always larger than that of the LR setting. The correlation depends on the signal-noise ratio $\sigma_x/\sigma_e$. In the noiseless case, $\sigma_e = 0$ and the expected correlation is $\sigma_x\sqrt{2/\pi}$, which is well known as the half-normal distribution. Similarly to analyzing RoLR for LR, based on Lemma 5, we can obtain the following performance guarantee for RoLR in solving classification problems.

**Theorem 2.** *Let $\hat{\beta}$ be the output of Algorithm 1, and $\beta^*$ be the optimal parameter minimizing the empirical risk. Suppose there are $n$ authentic samples generated by the model described by* (5). *Then we have, with large probability larger than $1 - 4\exp(-c_2 n/8)$,*

$$\|\hat{\beta} - \beta^*\|_2 \leq 2\lambda + 2(\lambda + 4 + 5\sqrt{\lambda})\sqrt{\frac{(\sigma_e^2 + \sigma_x^2)\pi p}{2\sigma_x^4 n}} + 8\lambda\sqrt{\frac{(\sigma_e^2 + \sigma_x^2)\pi}{2}}\sqrt{\frac{\log p}{n} + \frac{\log n}{n}}.$$

The proof of Theorem 2 is similar to that of Theorem 1. Also, similar to the LR case, based on the above parameter error bound, it is straightforward to obtain the empirical and population risk bounds of RoLR for classification. Due to the space limitation, here we only sketch how to obtain the risk bounds.

For the classification problem, the most natural loss function is the $0 - 1$ loss. However, $0 - 1$ loss function is non-convex, non-smooth, and we cannot get a non-trivial function value bound in terms of $\|\hat{\beta} - \beta^*\|$ as we did for the logistic loss function. Fortunately, several convex surrogate loss functions for $0-1$ loss have been proposed and achieve good classification performance, which include the hinge loss, exponential loss and logistic loss. These loss functions are all Lipschitz continuous and thus we can bound their empirical and then population risks as for logistic regression.

# 6 Simulations

In this section, we conduct simulations to verify the robustness of RoLR along with its applicability for robust binary classification. We compare RoLR with standard logistic regression which estimates the model parameter through maximizing the log-likelihood function.

We randomly generated the samples according to the model in Eqn. (1) for the logistic regression problem. In particular, we first sample the model parameter $\beta \sim \mathcal{N}(0, I_p)$ and normalize it as $\beta := \beta/\|\beta\|_2$. Here $p$ is the dimension of the parameter, which is also the dimension of samples. The samples are drawn i.i.d. from $x_i \sim \mathcal{N}(0, \Sigma_x)$ with $\Sigma_x = I_p$, and the Gaussian noise is sampled as $v_i \sim \mathcal{N}(0, \sigma_e)$. Then, the sample label $y_i$ is generated according to $\mathbb{P}\{y_i = +1\} = \tau(\langle \beta, x_i \rangle + v_i)$ for the LR case. For the classification case, the sample labels are generated by $y_i = \text{sign}(\langle \beta, x_i \rangle + v_i)$ and additional $n_t = 1,000$ authentic samples are generated for testing. The entries of outliers $x_o$ are i.i.d. random variables from uniform distribution $[-\sigma_o, \sigma_o]$ with $\sigma_o = 10$. The labels of outliers are generated by $y_o = \text{sign}(\langle -\beta, x_o \rangle)$. That is, outliers follow the model having opposite sign as inliers, which according to our experiment, is the most adversarial outlier model. The ratio of outliers over inliers is denoted as $\lambda = n_1/n$, where $n_1$ is the number of outliers and $n$ is the number of inliers. We fix $n = 1,000$ and the $\lambda$ varies from 0 to 1.2, with a step of 0.1.

We repeat the simulations under each outlier fraction setting for 10 times and plot the performance (including the average and the variance) of RoLR and ordinary LR versus the ratio of outliers to inliers in Figure 2. In particular, for the task of logistic regression, we measure the performance by the parameter prediction error $\|\hat{\beta} - \beta^*\|$. For classification, we use the classification error rate on test samples $- \#(\hat{y}_i \neq y_i)/n_t -$ as the performance measure. Here $\hat{y}_i = \text{sign}(\hat{\beta}^\top x_i)$ is the predicted label for sample $x_i$ and $y_i$ is the ground truth sample label. The results, shown in Figure 2,

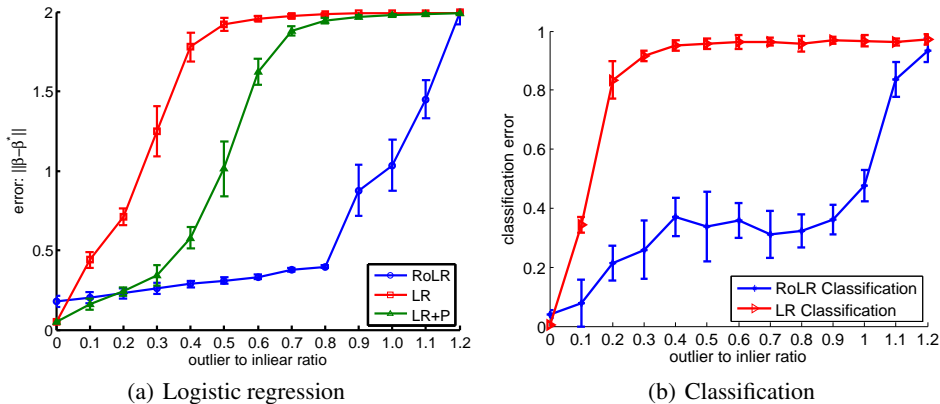

(a) Logistic regression　　　　　　　　　　(b) Classification

Figure 2: Performance comparison between RoLR, ordinary LR and LR with the thresholding pre-processing as in RoLR (LR+P) for (a) regression parameter estimation and (b) classification, under the setting of $\sigma_e = 0.5$, $\sigma_o = 10$, $p = 20$ and $n = 1,000$. The simulation is repeated for 10 times.

clearly demonstrate that RoLR performs much better than standard LR for both tasks. Even when the outlier fraction is small ($\lambda = 0.1$), RoLR already outperforms LR with a large margin. From Figure 2(a), we observe that when $\lambda \geq 0.3$, the parameter estimation error of LR reaches around 1.3, which is pretty unsatisfactory since simply outputting a trivial solution $\hat{\beta} = 0$ has an error of 1 (recall $\|\beta^*\|_2 = 1$). In contrast, RoLR guarantees the estimation error to be around 0.5, even though $\lambda = 0.8$, *i.e.*, around 45% of the samples are outliers. To see the role of preprocessing in RoLR, we also apply such preprocessing to LR and plot its performance as "LR+P" in the figure. It can be seen that the preprocessing step indeed helps remove certain outliers with large magnitudes. However, when the fraction of outliers increases to $\lambda = 0.5$, more outliers with smaller magnitudes than the pre-defined threshold enter the remained samples and increase the error of "LR+P" to be larger than 1. This demonstrates maximizing the correlation is more essential than the thresholding for the robustness gain of RoLR. From results for classification, shown in Figure 2(b), we observe that again from $\lambda = 0.2$, LR starts to breakdown. The classification error rate of LR achieves 0.8, which is even worse than random guess. In contrast, RoLR still achieves satisfactory classification performance with classification error rate around 0.4 even with $\lambda \to 1$. But when $\lambda > 1$, RoLR also breaks down as outliers dominate in the training samples.

When there is no outliers, with the same inliers ($n = 1 \times 10^3$ and $p = 20$), the error of LR in logistic regression estimation is 0.06 while the error of RoLR is 0.13. Such performance degradation in RoLR is due to that RoLR maximizes the linear correlation statistics instead of the likelihood as in LR in inferring the regression parameter. This is the price RoLR needs to pay for the robustness. We provide more investigations and also results for real large data in the supplementary material.

## 7 Conclusions

We investigated the problem of logistic regression (LR) under a practical case where the covariate matrix is adversarially corrupted. Standard LR methods were shown to fail in this case. We proposed a novel LR method, RoLR, to solve this issue. We theoretically and experimentally demonstrated that RoLR is robust to the covariate corruptions. Moreover, we devised a linear programming algorithm to solve RoLR, which is computationally efficient and can scale to large problems. We further applied RoLR to successfully learn classifiers from corrupted training samples.

### Acknowledgments

The work of H. Xu was partially supported by the Ministry of Education of Singapore through AcRF Tier Two grant R-265-000-443-112. The work of S. Mannor was partially funded by the Intel Collaborative Research Institute for Computational Intelligence (ICRI-CI) and by the Israel Science Foundation (ISF under contract 920/12).

## Footnotes

[1]It is defined as the percentage of corrupted points that can make the output of an algorithm arbitrarily bad.

[2]Here, the *parameter* means the sub-Gaussian norm of the random variable $Y$, $\|Y\|_{\psi_2} = \sup_{q \geq 1} q^{-1/2} (\mathbb{E}|Y|^q)^{1/q}$.

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
