[Supplementary Material]

# Robust Logistic Regression and Classification
## – Supplementary Material

**Jiashi Feng**
EECS Department & ICSI
UC Berkeley
jshfeng@berkeley.edu

**Huan Xu**
ME Department
National University of Singapore
mpexuh@nus.edu.sg

**Shie Mannor**
EE Department
Technion
shie@ee.technion.ac.il

**Shuicheng Yan**
ECE Department
National University of Singapore
eleyans@nus.edu.sg

## 1  Additional Experiments

We also evaluate the classification performance of ordinary LR and RoLR on the UCI SUSY dataset[1]. The SUSY dataset contains 5 million samples from 2 categories: background or particles. The feature dimension is 18. We randomly partition the dataset into training (including 3 million samples) and test (including 2 million samples) subsets for 10 times. The training data are corrupted by different fraction of outliers (ranging from 5% to 50%). Figure 1 shows the averaged classification error of LR and RoLR. As for the computational efficiency, it takes the ordinary LR 8.55 hours to learn the regression parameter in a single PC. In contrast, RoLR only costs 48 minutes for the parameter inference, whose time cost is less than $1/10$ of LR. Note that 0.45 is the classification error that can be achieved by trivially classifying all the samples to be positive.

Figure 1: Classification performance on SUSY dataset.

To see what RoLR needs to pay for the robustness, or more concretely the cost brought by optimizing the correlation instead of likelihood probability, we conduct several simulations over the data generated by the same experimental protocol as in the main body. Here, the sample dimension is varying from 10 to 200, and the number of samples is fixed as $n = 1 \times 10^3$. Table 1 gives the regression error of LR and RoLR as well as their computational time cost. Generally, RoLR suffers from

Table 1: Comparisons between LR and RoLR *without* any outliers. $n = 1 \times 10^3$.

| $p$ | 10 | 20 | 50 | 100 | 200 |
|---|---|---|---|---|---|
| LR error | 0.03 | 0.06 | 0.07 | 0.14 | 0.28 |
| RoLR error | 0.09 | 0.13 | 0.20 | 0.23 | 0.35 |
| LR time cost (sec.) | 0.43 | 0.42 | 0.41 | 0.44 | 0.43 |
| RoLR time cost (sec.) | 0.88 | 0.68 | 1.23 | 4.74 | 23.74 |

small error increase, compared with LR, when there are no outliers. Both the performance of LR and RoLR degrades along with the increasing of parameter dimension, as the number of the sample is fixed at a relatively small value. RoLR presents significant computational efficiency advantages over LR, especially for high-dimensional data.

## 2 Non-robustness of Ordinary Logistic Regression

In this section, we give the influence function of ordinary logistic regression to formally describe the robustness property of logistic regression. The *influence function* (IF) [2] of an estimator is an approximation to its behavior when the sample contains a small fraction $\varepsilon$ of identical outliers. It is defined as

$$\text{IF}_{\hat{\beta}}(x_0, F) = \lim_{\varepsilon \downarrow 0} \frac{\beta^*((1-\varepsilon)F + \varepsilon \delta_{x_0}) - \beta^*(F)}{\varepsilon} = \frac{\partial}{\partial \varepsilon} \beta^*((1-\varepsilon)F + \varepsilon \delta_{x_0}) \mid_{\varepsilon \downarrow 0},$$

where $\beta^*$ is the estimator, $F$ denotes the sample distribution, $\delta_{x_0}$ is the point-mass at $x_0$ and "$\downarrow$" stands for "limit from the right".

Logistic regression is not robust when the samples are corrupted by gross noise. Even one corrupted sample can manipulate the model estimation severely and make the result arbitrarily bad. To see this, recall the optimal solution to the above optimization problem $\beta^*$ satisfies,

$$\sum_{i=1}^{n} \frac{-y_i x_i}{1 + \exp(y_i \beta^{*\top} x_i)} = 0$$

We can calculate the influence function to show its non-robustness. The influence function of the MLE for the logistic regression model is [5]

$$\text{IF}(y, x, \beta) = M^{-1}(y - F(\langle \beta, x \rangle))x,$$

where $M = \mathbb{E}\left(F'(\langle \beta, x \rangle)xx^\top\right)$. Since the factor $(y - \tau(\langle \beta, x \rangle))$ is bounded, the only outliers that make this influence large are those such that $\|x\| \to \infty$. This kind of outliers make the MLE of $\beta$ tend to zero. Although the estimate remains bounded, we also say the logistic regression breaks down since the value of $\beta$ is manipulated by the outliers.

## 3 Objective function

In the following proofs, we define the objective function:

$$f_\beta(\hat{\beta}) = \frac{1}{n} \sum_{i=1}^{n} y_i \langle x_i, \hat{\beta} \rangle,$$

where $n$ is the number of inlier samples and $\beta$ is the groundtruth parameter of inlier generating model.

## 4 Proof of Lemma 3

*Proof.* We first note that if $X$ is a zero-mean sub-Gaussian variable with parameter $\sigma_x$, then the rescaled variable $X/\sigma_x$ is sub-Gaussian with parameter 1. Consequently, we may assume that $\sigma_x = 1$ without loss of generality, and with rescaling as necessary.

Let $g \triangleq \langle \beta^*, x \rangle + v$. Since $x \sim \mathcal{N}(0, I)$ is a standard Gaussian random vector and $\|\beta^*\| = 1$, the random variable $h \triangleq \langle \beta^*, x \rangle \sim \mathcal{N}(0, 1)$. Define the function $\theta(g) = \tanh(g/2) = \frac{1 - e^{-g}}{1 + e^{-g}}$, we have

$$\mathbb{E}[y|g] = \mathbb{P}\{y = +1|g\} - \mathbb{P}\{y = -1|g\} = \frac{1}{1 + e^{-g}} - \frac{e^{-g}}{1 + e^{-g}} = \frac{1 - e^{-g}}{1 + e^{-g}} = \theta(g).$$

Therefore,

$$\mathbb{E}y\langle \beta^*, x \rangle = \mathbb{E}yh = \mathbb{E}_{v,h}h\mathbb{E}_y[y|h, v] = \mathbb{E}_{v,h}\theta(h + v)h.$$

Denote $f(h)$ as the density function of $h$, which represents a normal distribution, *i.e.*, $f(h) = \frac{1}{\sqrt{2\pi}}e^{-\frac{h^2}{2}}$. Note that $f'(h) = -hf(h)$. We calculate the expectation of $\theta(h + v)h$ using integration by parts:

$$
\begin{aligned}
\mathbb{E}\theta(h + v)h &= \int\int \theta(h + v)hf(h)f(v)\mathrm{d}h\mathrm{d}v \\
&= -\int\int \theta(h + v)f'(h)f(v)\mathrm{d}h\mathrm{d}v \\
&= \int \left( \int \theta'(h + v)f(h)\mathrm{d}h - \theta(h + v)f(h)\Big|_{-\infty}^{+\infty} \right) f(v)\mathrm{d}v \\
&= \int\int \theta'(h + v)f(h)f(v)\mathrm{d}h\mathrm{d}v \\
&= \mathbb{E}_{h,v}\theta'(h + v) = \frac{1}{2}\mathbb{E}_g \operatorname{sech}^2(g/2).
\end{aligned}
$$

Here $g \triangleq h + v$. Since $h$ and $v$ are independent Gaussian random variable, $g$ is also a Gaussian random variable, and $g \sim \mathcal{N}(0, 1 + \sigma_e^2)$.

To further bound this quantity below, we can use the fact that $\operatorname{sech}^2(x)$ is an even and decreasing function for $x \geq 0$. This yields

$$
\begin{aligned}
\mathbb{E}\operatorname{sech}^2(g/2) &= \int_{-\infty}^{+\infty} \operatorname{sech}^2(g/2)f(g)\mathrm{d}g \\
&\geq \int_{-(1+\sigma_e^2)}^{1+\sigma_e^2} \operatorname{sech}^2(g/2)f(g)\mathrm{d}g \\
&\geq \mathbb{P}\left\{|g| \leq 1 + \sigma_e^2\right\} \cdot \operatorname{sech}^2\left(\frac{1 + \sigma_e^2}{2}\right) \\
&\geq \frac{2}{3}\operatorname{sech}^2\left(\frac{1 + \sigma_e^2}{2}\right).
\end{aligned}
$$

Hence,

$$\mathbb{E}y\langle \beta^*, x \rangle = \mathbb{E}\theta(h + v)h \geq \frac{1}{3}\operatorname{sech}^2\left(\frac{1 + \sigma_e^2}{2}\right).$$

Similarly, we can lower bound $\mathbb{E}\operatorname{sech}^2(g/2)$ by

$$
\begin{aligned}
\mathbb{E}\operatorname{sech}^2(g/2) &\leq \int_{-(\sigma_e^2+1)}^{\sigma_e^2+1} \operatorname{sech}^2(0)f(g)\mathrm{d}g + 2\int_{-\infty}^{-(\sigma_e^2+1)} \operatorname{sech}^2\left(\frac{1 + \sigma_e^2}{2}\right)f(g)\mathrm{d}g \\
&\leq \frac{2}{3} + \frac{1}{3}\operatorname{sech}^2\left(\frac{1 + \sigma_e^2}{2}\right).
\end{aligned}
$$

Therefore,

$$\mathbb{E}y\langle \beta^*, x \rangle = \mathbb{E}\theta(h + v)h \leq \frac{1}{3} + \frac{1}{6}\operatorname{sech}^2\left(\frac{1 + \sigma_e^2}{2}\right).$$

Taking the magnitude of the signal $x$, $\sigma_x$, into consideration, we have

$$\frac{\sigma_x}{3}\operatorname{sech}^2\left(\frac{1 + \sigma_e^2}{2}\right) \leq \sigma_x\mathbb{E}y\langle \beta^*, x/\sigma_x \rangle = \sigma_x\mathbb{E}\theta(h + v)h \leq \frac{\sigma_x}{3} + \frac{\sigma_x}{6}\operatorname{sech}^2\left(\frac{1 + \sigma_e^2}{2}\right).$$

$\square$

# 5  Proof of Lemma 4

*Proof.* Since the samples are i.i.d., we have

$$\mathbb{E}f_\beta(\hat{\beta}) = \frac{1}{n}\sum_{i=1}^{n}\mathbb{E}y_i\langle x_i, \hat{\beta}\rangle = \mathbb{E}y_1\langle x_1, \hat{\beta}\rangle.$$

Now we condition on $x_1$ to give

$$\mathbb{E}y_1\langle x_1, \hat{\beta}\rangle = \mathbb{E}\mathbb{E}[y_1\langle x_1, \hat{\beta}\rangle | x_1] = \mathbb{E}\theta(\langle x_1, \beta\rangle)\langle x_1, \hat{\beta}\rangle.$$

Note that $\langle x_1, \beta\rangle$ and $\langle x_1, \hat{\beta}\rangle$ are a pair of normal random variables with covariance $\langle \beta, \hat{\beta}\rangle$. Thus, by taking $g, h \in \mathcal{N}(0,1)$ to be independent, we may rewrite the above expectation as

$$\mathbb{E}\theta(g)\left(\langle \beta, \hat{\beta}\rangle g + (\|\hat{\beta}\|_2^2 - \langle\beta, \hat{\beta}\rangle^2)^{1/2}h\right) = \langle\beta, \hat{\beta}\rangle\mathbb{E}\theta(g)g = \eta\langle\beta, \hat{\beta}\rangle.$$

$\square$

# 6  Technical lemmas for proving Theorem 1

**Tools: symmetrization and Gaussian concentration**

**Lemma S.1** (Symmetrization [4]). *Let $\varepsilon_1, \varepsilon_2, \ldots, \varepsilon_n$ be independent Rademacher random variables. Let $K \subset B_2^p$. $K\backslash K = \{a - b, a, b \in K\}$. Then*

$$\mu := \mathbb{E}\sup_{z \in K\backslash K}|f_\beta(z) - \mathbb{E}f_\beta(z)| \leq 2\mathbb{E}\sup_{z\in K\backslash K}\frac{1}{n}\left|\sum_{i=1}^{n}\varepsilon_i y_i\langle x_i, z\rangle\right|. \tag{S.1}$$

*Furthermore, we have the deviation inequality*

$$\mathbb{P}\left\{\sup_{z\in K\backslash K}|f_\beta(z) - \mathbb{E}f_\beta(z)| \geq 2\mu + t\right\} \leq 4\mathbb{P}\left\{\sup_{z\in K\backslash K}\left|\sum_{i=1}^{n}\varepsilon_i y_i\langle x_i, z\rangle\right| > t/2\right\}. \tag{S.2}$$

In the above lemma, inequality (S.1) follows from the proof of Lemma 6.3 in [4], and inequality (S.2) is from Chapter 6.1 in [4].

**Theorem S.1** (Gaussian concentration inequality [6]). *Let $(G_x)_{x\in T}$ be a centered Gaussian process indexed by a finite set $T$. Then for every $r > 0$ one has*

$$\mathbb{P}\left\{\sup_{x\in T}G_x \geq \mathbb{E}\sup_{x\in T}G_x + r\right\} \leq \exp(-r^2/\sigma^2)$$

*where $\sigma^2 = \sup_{x\in T}\mathbb{E}G_x^2 < \infty$.*

A proof of this result is contained in [3].

This theorem can be extended to separable sets $T$ in metric spaces by an approximation argument. In particular, given a set $K \subseteq B_2^p$ and $r > 0$, the standard Gaussian random vector $g$ in $\mathbb{R}^p$ satisfies

$$\mathbb{P}\left\{\sup_{z\in K\backslash K}\langle g, z\rangle \geq \mathbb{E}\sup_{z\in K\backslash K}\langle g, z\rangle + r\right\} \leq \exp(-r^2/2). \tag{S.3}$$

Since the inlier samples obey the sub-Gaussian design and the parameter $\beta$ has a unit norm, the linear measure $\langle \beta, x_i\rangle$ is also a Gaussian random variable and possesses the following concentration property [6].

**Lemma S.2.** *For each $t > 0$, and $\beta \in S^{p-1}$, we have*

$$\mathbb{P}\left\{\left|\frac{1}{n}\sum_{i=1}^{n}y_i\langle\beta, x_i\rangle - \mathbb{E}y\langle\beta, x\rangle\right| > 4\sqrt{\frac{p}{n}} + t\right\} \leq 4\exp\left(-\frac{nt^2}{8}\right).$$

From Lemma S.2, we can easily obtain the following concentration result for the correlation differences.

**Lemma S.3.** *For each $t > 0$, we have*

$$\mathbb{P}\left\{\sup_{z \in S^{p-1} \backslash S^{p-1}} \left| \frac{1}{n} \sum_{i=1}^{n} y_i \langle z, x_i \rangle - \mathbb{E} y \langle z, x \rangle \right| \geq 5\sqrt{\frac{p}{n}} + t \right\} \leq 4 \exp\left(-\frac{nt^2}{8}\right).$$

*Here the set $S^{p-1} \backslash S^{p-1} = \{\beta - \beta' : \beta, \beta' \in S^{p-1}\}$.*

*Proof.* We apply the first part (S.1) of Symmetrization Lemma S.1. Note that since $x_i$ have symmetric distributions and $y_i \in \{-1, +1\}$, the random vectors $\varepsilon_i y_i x_i$ has the same (iid) distribution as $x_i$. Using the rotational invariance and the symmetry of the Gaussian distribution, we can represent the right hand side of (S.1) as

$$\sup_{z \in K \backslash K} \frac{1}{n} \left| \sum_{i=1}^{n} \varepsilon_i y_i \langle x_i, z \rangle \right| \stackrel{dist}{=} \sup_{z \in K \backslash K} \frac{1}{n} \left| \sum_{i=1}^{n} \langle x_i, z \rangle \right| \stackrel{dist}{=} \frac{1}{\sqrt{n}} \sup_{z \in K \backslash K} |\langle g, z \rangle| = \frac{1}{\sqrt{n}} \sup_{z \in K - K} \langle g, z \rangle,$$

(S.4)

where $\stackrel{dist}{=}$ signifies the equality in distribution. So taking the expectation in (S.1) we obtain

$$\mathbb{E} \sup_{z \in K \backslash K} |f_\beta(z) - \mathbb{E} f_\beta(z)| \leq \frac{2}{\sqrt{n}} \mathbb{E} \sup_{z \in K \backslash K} \langle g, z \rangle. \tag{S.5}$$

To supplement this expectation bound with a deviation inequality, we use the second part (S.2) of Symmetrization Lemma S.1 along with (S.4) and (S.5). This yields

$$\mathbb{P}\left\{\sup_{z \in K \backslash K} |f_\beta(z) - \mathbb{E} f_\beta(z)| \geq \frac{4}{\sqrt{n}} \mathbb{E} \sup_{z \in K \backslash K} \langle g, z \rangle + t \right\} \leq 4\mathbb{P}\left\{\frac{1}{\sqrt{n}} \sup_{z \in K \backslash K} \langle g, z \rangle > t/2 \right\}.$$

Now it remains to use the Gaussian concentration inequality (S.3) with $r = t\sqrt{m}/2$. The proof is complete. □

## 7 Proof of Theorem 1

*Proof.* In Algorithm 1, we actually minimize the empirical loss over trimmed samples. Let $\mathcal{I}$ denote the index set of the trimmed inliers, and let $\mathcal{O}$ denote the index set of the remained outliers. Note that $n_2 := |\mathcal{I}| = |\mathcal{O}| \leq n_1$. Let $\mathcal{I}'$ denote the index set of the remained inliers.

By the definition of Algorithm 1, for the solution output $\hat{\beta}$, we have

$$\sum_{i=1}^{n} y_i \langle \hat{\beta}, x_i \rangle - \sum_{i \in \mathcal{I}} y_i \langle \hat{\beta}, x_i \rangle + \sum_{i \in \mathcal{O}} y_i \langle \hat{\beta}, x_i \rangle \geq \sum_{i=1}^{n} y_i \langle \beta^*, x_i \rangle - \sum_{i \in \mathcal{I}} y_i \langle \beta^*, x_i \rangle + \sum_{i \in \mathcal{O}} y_i \langle \beta^*, x_i \rangle.$$

Thus,

$$\sum_{i=1}^{n} y_i \langle \beta^*, x_i \rangle - \sum_{i=1}^{n} y_i \langle \hat{\beta}, x_i \rangle \leq \sum_{i \in \mathcal{I}} y_i \langle \beta^*, x_i \rangle - \sum_{i \in \mathcal{I}} y_i \langle \hat{\beta}, x_i \rangle + \sum_{i \in \mathcal{O}} y_i \langle \hat{\beta}, x_i \rangle - \sum_{i \in \mathcal{O}} y_i \langle \beta^*, x_i \rangle.$$

(S.6)

According to Algorithm 1, the trimmed samples have smaller value of $y_i \langle \hat{\beta}, x_i \rangle$ (larger empirical loss) than remained samples. Therefore, $-\sum_{i \in \mathcal{I}} y_i \langle \hat{\beta}, x_i \rangle + \sum_{i \in \mathcal{O}} y_i \langle \hat{\beta}, x_i \rangle < 0$. We get

$$\frac{1}{n} \sum_{i=1}^{n} y_i \langle \beta^*, x_i \rangle - \frac{1}{n} \sum_{i=1}^{n} y_i \langle \hat{\beta}, x_i \rangle \leq \frac{1}{n} \sum_{i \in \mathcal{I}} y_i \langle \beta^*, x_i \rangle - \frac{1}{n} \sum_{i \in \mathcal{O}} y_i \langle \beta^*, x_i \rangle.$$

Again from the definition of the algorithm, we have

$$\frac{1}{n}\sum_{i=1}^{n} y_i\langle\beta^*, x_i\rangle - \frac{1}{n}\sum_{i=1}^{n} y_i\langle\hat{\beta}, x_i\rangle$$

$$\leq \quad \frac{1}{n}\sum_{i\in\mathcal{I}} y_i\langle\beta^*, x_i\rangle - \frac{1}{n}\sum_{i\in\mathcal{O}} y_i\langle\beta^*, x_i\rangle$$

$$\leq \quad \frac{1}{n}\sum_{i\in\mathcal{I}} y_i\langle\beta^*, x_i\rangle + \frac{1}{n}\sum_{i\in\mathcal{O}} |\langle\beta^*, x_i\rangle|$$

$$\overset{(a)}{\leq} \quad \frac{1}{n}\sum_{i\in\mathcal{I}} y_i\langle\beta^*, x_i\rangle + \frac{1}{n}\sum_{i\in\mathcal{O}} \|x_i\|$$

$$\overset{(b)}{\leq} \quad \frac{1}{n}\sum_{i\in\mathcal{I}} y_i\langle\beta^*, x_i\rangle + \frac{4n_1}{n}\sigma_x^2\sqrt{\frac{\log p}{n} + \frac{\log n}{n}}.$$

The inequality $(a)$ is from the fact that $|\langle\beta^*, x_i\rangle| \leq \|\beta^*\|\|x_i\| = \|x_i\|$, and the inequality $(b)$ is from the definition of the algorithm.

Applying the above expectation (Lemma 3) and concentration (Lemma S.2) results of the function $y_i\langle\beta^*, x_i\rangle$ gives

$$\frac{1}{n_2}\sum_{i\in\mathcal{I}} y_i\langle\beta^*, x_i\rangle \leq \mathbb{E}\left[y_i\langle\beta^*, x_i\rangle\right] + 5\sqrt{\frac{p}{n_2}} \leq \varphi^-(\sigma_e^2, \sigma_x^2) + 5\sqrt{\frac{p}{n_2}},$$

with probability $1 - 4\exp(-cn/8)$, where $n_2 = |\mathcal{I}| = |\mathcal{O}| \leq n_1$. Thus we have,

$$\frac{1}{n}\sum_{i=1}^{n} y_i\langle\beta^*, x_i\rangle - \frac{1}{n}\sum_{i=1}^{n} y_i\langle\hat{\beta}, x_i\rangle$$

$$\leq \quad \frac{n_2}{n}\left(\frac{1}{n_2}\sum_{i\in\mathcal{I}} y_i\langle\beta^*, x_i\rangle\right) + \frac{4n_1}{n}\sigma_x^2\sqrt{\frac{\log p}{n} + \frac{\log n}{n}}$$

$$\leq \quad \frac{n_2}{n}\left(\varphi^-(\sigma_e^2, \sigma_x^2) + 5\sqrt{\frac{p}{n_2}}\right) + \frac{4n_1}{n}\sigma_x^2\sqrt{\frac{\log p}{n} + \frac{\log n}{n}}$$

$$\leq \quad \frac{n_1}{n}\left(\varphi^-(\sigma_e^2, \sigma_x^2) + 5\sqrt{\frac{p}{n_1}}\right) + \frac{4n_1}{n}\sigma_x^2\sqrt{\frac{\log p}{n} + \frac{\log n}{n}} \tag{S.7}$$

$$= \quad \lambda\left(\varphi^-(\sigma_e^2, \sigma_x^2) + 5\sqrt{\frac{p}{n_1}}\right) + 4\lambda\sigma_x^2\sqrt{\frac{\log p}{n} + \frac{\log n}{n}}. \tag{S.8}$$

where we apply the fact that $n_2 \leq n_1$ and $\lambda \triangleq \frac{n_1}{n}$. According to Lemma 4 and Lemma S.3,

$$\|\hat{\beta} - \beta^*\|$$

$$\leq \quad \frac{2}{\varphi^+(\sigma_e^2, \sigma_x^2)}\mathbb{E}\left[y\langle x, \beta^*\rangle - y\langle x, \hat{\beta}\rangle\right]$$

$$\leq \quad \frac{2}{\varphi^+(\sigma_e^2, \sigma_x^2)}\left(\frac{1}{n}\sum_{i=1}^{n} y_i\langle\beta^*, x_i\rangle - \frac{1}{n}\sum_{i=1}^{n} y_i\langle\hat{\beta}, x_i\rangle + (\lambda + 4)\sqrt{\frac{p}{n}}\right),$$

holds with probability larger than $1 - 4\exp(-cn/8)$. Substituting (S.7) into above equality yields

$$\|\hat{\beta} - \beta^*\|$$

$$\leq \quad \frac{2}{\varphi^+(\sigma_e^2, \sigma_x^2)}\left\{\lambda\left(\varphi^-(\sigma_e^2, \sigma_x^2) + 5\sqrt{\frac{p}{n_1}}\right) + 4\lambda\sigma_x^2\sqrt{\frac{\log p}{n} + \frac{\log n}{n}} + (\lambda + 4)\sqrt{\frac{p}{n}}\right\}$$

$$= \quad 2\lambda\frac{\varphi^-(\sigma_e^2, \sigma_x^2)}{\varphi^+(\sigma_e^2, \sigma_x^2)} + \frac{2(\lambda + 4 + 5\sqrt{\lambda})}{\varphi^+(\sigma_e^2, \sigma_x^2)}\sqrt{\frac{p}{n}} + \frac{8\lambda}{\varphi^+(\sigma_e^2, \sigma_x^2)}\sigma_x^2\sqrt{\frac{\log p}{n} + \frac{\log n}{n}}.$$

$\square$

# 8 Proof of Corollary 1

*Proof.* We first provide a bound of the empirical risk of RoLR as follows,

$$|R_{\text{emp}}(\hat{\beta}) - R_{\text{emp}}(\beta^*)|$$

$$= \frac{1}{n}\sum_{i=1}^{n}\left(\frac{1}{1+\exp\{-y_i\langle\hat{\beta},x_i\rangle\}} - \frac{1}{1+\exp\{-y_i\langle\beta^*,x_i\rangle\}}\right)$$

$$\overset{(a)}{\leq} \frac{L}{n}\sum_{i=1}^{n}|y_i\langle\hat{\beta},x_i\rangle - y_i\langle\beta^*,x_i\rangle|$$

$$= \frac{L}{n}\sum_{i=1}^{n}|\langle\hat{\beta}-\beta^*,x_i\rangle|$$

$$\leq L\|\hat{\beta}-\beta^*\|\frac{1}{n}\sum_{i=1}^{n}\|x_i\|.$$

$$\overset{(b)}{\leq} L\|\hat{\beta}-\beta^*\|4\sigma_x\sqrt{(\log n + \log p)/n}. \tag{S.9}$$

The inequality $(a)$ is from the fact that function $\tau(z) = 1/(1+e^{-z})$ is Lipschitz continuous with Lipschitz constant $L = 1$:

$$|\tau(z) - \tau(z')| \leq |z - z'|,$$

and the inequality $(b)$ is from Lemma 2 that the magnitude of inliers is upper bounded.

Then applying the result in Theorem 1 directly gives the following empirical risk bound for RoLR. $\square$

# 9 Proof of Corollary 2

*Proof.* Let $\beta^*$ denote the optimal parameter can be learned from authentic training samples, and $\hat{\beta}$ denote the output of the Algorithm 1, we aim to bound the difference of their population risk:

$$|R(\hat{\beta}) - R(\beta^*)|$$

$$= |R(\hat{\beta}) - R_{\text{emp}}(\hat{\beta}) + R_{\text{emp}}(\hat{\beta}) - R_{\text{emp}}(\beta^*) + R_{\text{emp}}(\beta^*) - R(\beta^*)|$$

$$\leq |R(\hat{\beta}) - R_{\text{emp}}(\hat{\beta})| + |R_{\text{emp}}(\hat{\beta}) - R_{\text{emp}}(\beta^*)| + |R_{\text{emp}}(\beta^*) - R(\beta^*)|.$$

Suppose the parameter space $\mathcal{B} \ni \beta$ has finite VC dimension $d$, we can apply the uniform convergence bound here to bound the *generalization risk* (*i.e.*, the difference between the population risk and empirical risk for a specific parameter $\beta$) in the above first and third term [1], *i.e.*,

$$|R(\hat{\beta}) - R_{\text{emp}}(\hat{\beta})| \leq c_3\sqrt{\frac{d + \ln(1/\delta)}{n}},$$

$$|R(\beta^*) - R_{\text{emp}}(\beta^*)| \leq c_4\sqrt{\frac{d + \ln(1/\delta)}{n}},$$

with probability at least $(1 - \delta)$. Here $c_3$ and $c_4$ are constants. Then we get the population risk bound. $\square$

# 10 Proof of Lemma 5

*Proof.* The proof is similar to the one for Lemma 3.

Let $g \triangleq \langle\beta^*,x\rangle + v$. Since $x \sim \mathcal{N}(0,\sigma_x^2 I)$ is a standard Gaussian random vector and $\|\beta^*\| = 1$, the random variable $h \triangleq \langle\beta^*,x\rangle \sim \mathcal{N}(0,\sigma_x^2)$. Thus the random variable $g = h + v$ is also a Gaussian random variable: $g \sim \mathcal{N}(0,\sigma_x^2 + \sigma_e^2)$.

Recall $y = \text{sign}(\langle\beta^*,x\rangle + v)$, and we calculate the expectation of the correlation $y\langle\beta^*,x\rangle$ as follows,

$$\mathbb{E}y\langle\beta^*,x\rangle = \mathbb{E}\text{sign}(v+h)h.$$

Recall that $h$ and $v$ are independent, we can also calculate the above expectation via integration by parts:

$$
\begin{aligned}
&\mathbb{E}\mathrm{sign}(v+h)h \\
=\ & \int\int \mathrm{sign}(v+h)hf(h)f(v)\mathrm{d}h\mathrm{d}v \\
=\ & -\sigma_x^2 \int \left(\int \mathrm{sign}(v+h)f(h)'\mathrm{d}h\right)f(v)\mathrm{d}v \\
=\ & -\sigma_x^2 \int \left(-\int_{-\infty}^{-v} f(h)'\mathrm{d}h + \int_{-v}^{+\infty} f(h)'\mathrm{d}h\right)f(v)\mathrm{d}v \\
=\ & \sigma_x^2 \int \left(1 - 2\int_{-v}^{+\infty} f(h)'\mathrm{d}h\right)f(v)\mathrm{d}v \\
=\ & \sigma_x^2 \int \left(1 - 2f(h)|_{-v}^{+\infty}\right)f(v)\mathrm{d}v \\
=\ & \sigma_x^2 \sqrt{\frac{2}{\pi(\sigma_x^2+\sigma_v^2)}}
\end{aligned}
$$

$\square$

## 11  Proof of Theorem 2

*Proof.* The proof of the parameter estimation bound in Theorem 2 is similar to the proof of Theorem 1 as the optimization algorithms are the same. Thus, based on the definition of Algorithm 1 and the distribution assumption of the samples, we have a result for the classification case which is similar to Eqn. (S.8):

$$
\frac{1}{n}\sum_{i=1}^n y_i\langle\beta^*,x_i\rangle - \frac{1}{n}\sum_{i=1}^n y_i\langle\hat{\beta},x_i\rangle \le \lambda\left(\mathbb{E}y_i\langle\beta^*,x_i\rangle + 5\sqrt{\frac{p}{n_1}}\right) + 4\lambda\sigma_x^2\sqrt{\frac{\log p}{n} + \frac{\log n}{n}}.
$$

According to Lemma 5, we have $\mathbb{E}y_i\langle\beta^*,x_i\rangle = \sigma_x^2\sqrt{\frac{2}{\pi(\sigma_x^2+\sigma_v^2)}}$. Substituting it into the above equation yields:

$$
\frac{1}{n}\sum_{i=1}^n y_i\langle\beta^*,x_i\rangle - \frac{1}{n}\sum_{i=1}^n y_i\langle\hat{\beta},x_i\rangle \le \lambda\left(\sigma_x^2\sqrt{\frac{2}{\pi(\sigma_x^2+\sigma_v^2)}} + 5\sqrt{\frac{p}{n_1}}\right) + 4\lambda\sigma_x^2\sqrt{\frac{\log p}{n} + \frac{\log n}{n}}.
$$

(S.10)

According to Lemma 4 and Lemma S.3,

$$
\begin{aligned}
\|\hat{\beta}-\beta^*\| &\le \frac{\sqrt{2\pi(\sigma_x^2+\sigma_e^2)}}{\sigma_x^2}\mathbb{E}\left[y\langle x,\beta^*\rangle - y\langle x,\hat{\beta}\rangle\right] \\
&\le \frac{\sqrt{2\pi(\sigma_x^2+\sigma_e^2)}}{\sigma_x^2}\left(\frac{1}{n}\sum_{i=1}^n y_i\langle\beta^*,x_i\rangle - \frac{1}{n}\sum_{i=1}^n y_i\langle\hat{\beta},x_i\rangle + (\lambda+4)\sqrt{\frac{p}{n}}\right),
\end{aligned}
$$

holds with probability larger than $1 - 4\exp(-cn/8)$. Substituting (S.10) into above equality yields

$$
\begin{aligned}
&\|\hat{\beta}-\beta^*\| \\
\le\ & 2\lambda + 2(\lambda+4+5\sqrt{\lambda})\sqrt{\frac{(\sigma_e^2+\sigma_x^2)\pi p}{2\sigma_x^4 n}} + 8\lambda\sqrt{\frac{(\sigma_e^2+\sigma_x^2)\pi}{2}}\sqrt{\frac{\log p}{n}+\frac{\log n}{n}}.
\end{aligned}
$$

$\square$

## Footnotes

[1] https://archive.ics.uci.edu/ml/datasets/SUSY