[Reviews · NeurIPS 2014]

Submitted by Assigned_Reviewer_39

The authors consider logistic regression when the data are contaminated with outliers. No assumptions are made on the outliers, they need not even follow any distribution, which can happen for example when a video is corrupted by sensor errors.

The authors provide an innovative algorithm, RoLR, and provide performance guarantees. The algorithm is simple to describe but uses sophisticated ideas, and the mathematics behind the performance guarantees is impressive. The simulations are promising.
Specific comments:
L135: Why does $\beta^*$ have unit length?
Line 157: X_i is p-dimensional, so how is the square and the absolute value defined?

The paper is of high quality and clearly written. It is to be seen how the method will perform for real data, but regardless of this the theoretical advance is important.

Summary: The paper is innovative and the mathematics is of high quality.

Submitted by Assigned_Reviewer_44

The manuscript proposes a new method for robust logistic regression with a focus on dealing with outliers with high leverage. Outliers are assumed to come from an arbitrary and unknown distribution, but the number of outliers is assumed to be known ahead of time, which is a theoretically convenient, but practically slightly more troubling assumption.

The method is based on maximising the sum of y*x'beta, but only summed over the observations which contribute in absolute terms the least to the objective function, thus outliers with large leverage are excluded. This can be elegantly translated into a linear programming problem.

The authors derive some risk bounds. In common with most of these bounds, they are probably too loose to have a practical quantitative use, but are useful for qualitative interpretation.

The manuscript contains a small simulation study in which the proposed method compares very favourably with classical logistic regression.

A few more detailed comments:

- Why do you need the preprocessing step? Observations with large ||x|| should also be among the observations with large y*x'beta and thus omitted by the algorithm. Also T is decreasing in n (and tends to 0 in the limit), so the more data we have, the more observations are thrown out by the preprocessing. For example, what proportion of observations is removed by the preprocessing step in the simulation?
Also, I am surprised that T is not chosen based on n or lambda.

- On a related point, it might be worth looking at how well logistic regression does after the same preprocessing is carried out.

- Given the preprocessing and the set-up of the method, situations where (like in the simulation shown) the outliers have much larger variance in the covariates are quite favourable to the proposed method (they are very likely to trigger the preprocessing criterion or lead to a large contribution to the objective function and are thus omitted). What happens is sigma_o is decreased, so that outliers look less like obvious outliers and are more likely to sneak into the n smallest contributions (and thus not excluded by the method)? Will the method perform better or worse?

- One important question is how much worse the method performs when compared to logistic regression when there are no outliers. The very left end of figure 2 suggests no big difference in terms of estimating beta, but a more substantial difference when looking at the misclassification rate. Essentially, what price do we have to pay for robustness?

- In practice, how do I choose n? How sensitive is the method to the choice of n?

The manuscript is well written and clearly structured.
Summary: The manuscript, which is well written, proposes a new method for performing robust logistic regression, which is essentially based on spotting observation with large leverage. The bit that impressed me the most is how the authors turn the problem of finding the observation contributing the least to the objective function into a linear programming problem.

Submitted by Assigned_Reviewer_45

This paper introduce a robust logistic regression under arbitrary outliers. As the title of the paper implies, the method is robust to outliers, and this point is investigated in terms of theory as well as experiments.

The theoretical result to outliers (theorem 1 and remark 1) is impressive. The estimation algorithm is also nice, which consists of just thresholding and linear programming. However, in practical points of view, the method includes some weakness, which is that the user has to set the number of outliers in advance. This makes the method weaker in practice because it is difficult to know the number in advance. If author(s) discussed ideas to overcome this problem, the paper would be better. In addition, the author(s) should mention how the performance varies if the incorrect number of outliers were set in the method.
Summary: The paper is well-written, and some theoretical result is impressive. I think that this work has an impact on machine learning community.
Author Feedback
Author rebuttal: We thank all the reviewers for their positive and constructive comments. Here we clarify that we do not need to know the exact number of outliers in applying the RoLR algorithm. An upper bound on the number of outliers is sufficient, or we can simply set the outlier number as half of the total number of the samples. Followings are point-to-point replies to the specific comments.

To reviewer 39

Q1. L135: Why does $\beta^*$ have unit length?

A1. In RoLR, we resort to maximizing the correlation $yi*\beta^\top xi$ for the estimate of the LR parameter \beta. Without constraints on the magnitude of \beta, the value of the objective function can be arbitrarily large. Thus, we add such a constraint on \beta. Note that this constraint does not damage the generality of RoLR. Since LR only concerns the inner product between \beta and xi, we can always scale xi to make the magnitude of the optimal \beta less than 1.

Q2. Line 157: X_i is p-dimensional, so how is the square and the absolute value defined?

A2. Sorry for the typo. Here X_i is a random scalar variable. The elements in a p-dimensional vector X_i are assumed to be i.i.d. Thus we define the variance for each element individually.

To reviewer 44

Q1. The number of outliers is assumed to be known ahead of time. In practice, how do I choose n? How sensitive is the method to the choice of n?

A1. We apologize for causing the confusion. Actually, we do not necessarily require the exact number of outliers or n. An upper bound on the number of outliers (which can be simply set as half of the total number of samples) is sufficient in the implementation. Of course, if we set n (number of inliers) too small, we will remove many useful inliers, which will increase the estimation error, as shown in Theorem 1.

Q2. Why do you need the preprocessing step? what proportion of observations is removed by the preprocessing step in the simulation? I am surprised that T is not chosen based on n or lambda.

A2. The preprocessing step is for upper bounding the magnitude of the remained outliers. The threshold value of T is chosen based on the sub-Gaussian distribution assumption on inliers. Most of the inliers have the magnitude smaller than T, due to their concentration property (Definition 1 and Lemma 2). Thus, applying such a preprocessing, we can upper bound the outliers and meanwhile preserve most of the inliers.

Basically, T is proportional to the expected magnitude of inlier samples, and thus T is proportional to the variance of inliers. In this work, we adopt the sub-Gaussian assumption which is standard in previous works [4, 13]. With such an assumption, the variance of the inliers decreases along with the increase of n. This is why the value of T also decreases with n. If we use other distributional assumptions, we can simply change the value of T according to the corresponding variance or tail estimates. The error bound eventually depends on p and the variance.

In the simulations, we found that actually none of the inliers were removed in the pre-processing as T is relatively larger than the magnitude of the inliers. We will state this point in the revision.

T is only used for removing outliers with extremely large magnitude. Thus T does not depend on lambda. The robustness of the RoLR essentially roots in maximizing the sorted and trimmed statistics (Equation 2), which depends on n or lambda.

Q3. It might be worth looking at how well logistic regression does after the same preprocessing is carried out.

A3. Thanks for the suggestion. As mentioned in A2, no samples were actually removed in the experiments since the threshold value T is relatively large. We will add the simulation results for such a case where the pre-processing also applies for the traditional LR.

Q4. What happens is sigma_o is decreased? Will the method perform better or worse?

A4. If sigma_o is decreased, the effect of the outliers on the parameter estimation will become small. It is not easy to give a deterministic answer to whether the final performance will be better or worse. It depends on how many outliers enter the n smallest contributions and how large the magnitude of those outliers. If the number of outliers is large however the magnitudes are small, it is possible that the final performance can be better.

Q5. How much worse the method performs when compared to logistic regression when there are no outliers. What price do we have to pay for robustness?

A5. Thanks for the suggestion. We will add experimental comparisons between LR and RoLR when there are no outliers. Basically, the performance degradation depends on the correlation between labels and samples’ response on the ground truth model: yi*{\beta^*}^\top xi. We can get this from Theorem 1 if we set \lambda = 0. The estimation error |\hat{beta}-\beta^*|is proportional to $8/\phi^+ \sqrt{p/n}$. Here \phi^+ is a lower bound on the expectation of y*\beta^\top x. When the correlation is strong, such as in binary classification, the estimation error could be small. Such estimation error is the cost for RoLR to maximize the correlation instead of optimizing the original likelihood function of LR, for the sake of robustness.

To reviewer 45

Q1. The user has to set the number of outliers in advance. The author(s) should mention how the performance varies if the incorrect number of outliers were set.

A1. Actually, we do not need to know the exact number of outliers in applying the RoLR algorithm. An upper bound on the number of outliers is sufficient, or we can simply set the outlier number as 0.5 of the total number of the samples. If we set the number of outliers too conservative (a quite small n), we will discard useful inliers. This will increase the estimator error at the order of \sqrt{1/n}, as given in Theorem 1. In the revision, we will add the discussions about choosing the number of outliers and thanks for the suggestion.